# Learning to Prompt Your Domain for Federated Vision-Language Models

**Guoyizhe Wei**
*Johns Hopkins University*

**Feng Wang**
*Johns Hopkins University*

**Anshul Shah**
*Johns Hopkins University*

**Rama Chellappa**
*Johns Hopkins University*

**Reviewed on OpenReview:** *https://openreview.net/forum?id=OS7zPOZjr3*

## Abstract

The prompt tuning paradigm, with its great advantages of low parameter count and stable training, has recently inspired numerous applications of CLIP-like vision-language models in federated learning. However, in this work, we posit that under significant domain gaps across federated participants, prompt-based CLIP may easily collapse to non-optimal solutions due to the neglect of domain-aware knowledge. We present a novel prompt tuning method, termed ADAPT, to address this issue by learning both intra- and inter-domain prompts. Specifically, we assign each federated participant a domain-specific prompt and use the image's visual features as a condition to guide the generation of language features, with the underlying idea that the prompted CLIP should detect the input image's domain correspondence before making the prediction of its category. Extensive experiments demonstrate ADAPT's significant efficiency and effectiveness in federated learning. For example, by learning and sharing only 0.35M parameters, ADAPT attains a 69.8% average accuracy over the six domains of DomainNet, which improves the original CLIP accuracy by 16.2%.

## 1 Introduction

Contrastive Language Image Pretraining (CLIP) (Radford et al., 2021) has recently been proven to be a powerful framework for multi-modal representation learning. By connecting the latent spaces of visual and textual inputs, CLIP offers a convenient approach of open-vocabulary inference for various downstream tasks. Building upon CLIP, the prompt learning technique, which freezes the encoders and introduces learnable tokens at the input side, can help CLIP adapt to downstream domains with minimal cost. Compared to the traditional finetuning paradigm, prompt learning approaches can produce competitive results with much fewer learnable parameters (*e.g.*, 0.1% of encoder parameters). This significant advantage in parameter efficiency has motivated numerous recent works to explore the potential applications of prompt-based CLIP in Federated Learning, where communication efficiency and training stability play important roles (Lu et al., 2023; Guo et al., 2023; Yang et al., 2023; Su et al., 2024; Li et al., 2024).

However, in this work, we identify that prompt learning approaches are sensitive in processing domain information, with prompts learned from a specific domain often struggling to transfer effectively to new data with significantly different features. This issue has greatly limited the applications of prompt tuning in federated learning scenarios. As in federated learning, one of the primary objectives is to learn from

heterogeneous data, where this heterogeneity may manifest as domain gaps when feature distributions become significantly diverse. To thoroughly examine the potential of prompt tuning in federated learning, we consider a challenging yet realistic scenario in which we assume there are multiple participants intending to train a shared model with their local data originating from different domains. Following the previous practice (Peng et al., 2020), we formulate this scenario using domain-aware datasets like DomainNet (Peng et al., 2019), where there are labeled images sourced from six distinct domains with different styles like real-world, paining, and sketch. Due to large diversity in input, conventional domain-agnostic federated learning approaches often struggle to generalize well in this scenario.

Under this challenging scenario, we find most parameter-efficient federated learning methods—which previously worked well in simpler heterogeneous settings with label-wise non-i.i.d. fail to produce favorable results. This observation suggests that, federated learning tasks involving significant domain gaps cannot be addressed solely by collaborative training methods; instead, they require the careful incorporation of domain awareness into the model. To this end, we propose a new federated learning framework termed ADAPT (Feder**A**ted **D**omain-**A**ware **P**rompt **T**uning), to enhance prompt tuning's ability to handle highly heterogeneous data in federated learning by incorporating domain-wise knowledge. Specifically, ADAPT is built upon a pretrained CLIP model and consists of three key components: 1) visual and 2) textual prompts are set up for each domain, where each visual prompt denotes a single learnable token appended to patch-embedded image tokens, and each text prompt consists of several learnable tokens representing textual descriptions indicative of each domain's style information; and 3) a vision-language coupling module done by AdaLN (Perez et al., 2018). We highlight that ADAPT effectively incorporates domain knowledge, achieving significantly superior predictive performance over traditional prompt tuning baselines in domain-aware federated learning tasks. Our contributions can be summarized as follows:

- **Domain-aware prompt learning**. We assign domain-specific textual prompts to each federated participant, enabling the model to make predictions by input images' corresponding domain information, which effectively addresses the widespread issue of domain gaps in federated learning. Our experimental results demonstrate a significant performance improvement with this design: based on a pretrained CLIP model equipped with a ViT-Base image encoder, ADAPT achieves an average accuracy of 69.8% on DomainNet, which significantly outperforms the zero-shot CLIP's 53.6%, the basic prompt learning's 63.2%, and PedProx's (Li et al., 2020b) 55.3% with a same image encoder.

- **Efficient communication**. ADAPT requires training and sharing only a small fraction of parameters, significantly reducing the communication overhead in federated learning. For instance, in our domain-aware federated learning experiments, ADAPT, with just 0.35M trainable parameters, achieved state-of-the-art performance on the DomainNet (Peng et al., 2019), Office-Home (Venkateswara et al., 2017), and PACS (Li et al., 2017) datasets.

- **Superior privacy preservation**. Due to the reduction in the number of trainable parameters in ADAPT, traditional federated learning attack algorithms (Zhu et al., 2019; Geiping et al., 2020) struggle to reconstruct the local data of participants from model gradients. Additionally, the learnable prompts themselves do not leak customer privacy—we have attempted to decode our prompts but found it difficult to extract any interpretable information from them.

## 2 Related Work

**Federated learning** was first introduced in the Federated Averaging (FedAvg) paper (McMahan et al., 2017), addressing machine learning problems with massively distributed private data. To enhance the learning potential of FedAvg, FedProx (Li et al., 2020b) adds a $\ell_2$ regularization term into the FedAvg's objective. Following FedAvg's success, many follow-up works improve federated learning in terms of privacy-preserving potentials (Wei et al., 2020; Truex et al., 2019), robustness to heterogeneous data (Karimireddy et al., 2020; Li et al., 2019), communication efficiency (Konečný et al., 2016; Sattler et al., 2019), and compatibility to model architectures (Li et al., 2020a; Qu et al., 2022). Unlike general federated learning methods that simulate non-i.i.d. data by partitioning datasets in the label space, many recent works consider federated learning in the more realistic context of domain adaptation (Yao et al., 2022a; Shenaj et al., 2023; Peng

et al., 2020). Recently, based on advances in multi-modal contrastive learning (Radford et al., 2021), various works develop CLIP-based federated learning methods (Lu et al., 2023; Li et al., 2024; Su et al., 2024; Qiu et al., 2024). For example, FedCLIP (Lu et al., 2023) leverages a pre-trained CLIP model with an additional adaptor layer for federated training. FedOPT (Li et al., 2024) trains both global and local prompts, while FedAPT (Su et al., 2024) creates global prompts from local ones. FedTPG (Qiu et al., 2024) develops a global prompt generator that transforms class names into prompt vectors. Similarly, PromptFL (Guo et al., 2023) employs simple prompt learning techniques to enhance federated optimization.

**Vision-language models.** Following the success of contrastive pre-training in visual modality (He et al., 2020; Chen et al., 2020; Grill et al., 2020; Caron et al., 2021; Chen & He, 2021; Chen et al., 2021), multi-modal contrastive pre-training has become a common paradigm in recent years as well. A representative work is CLIP (Radford et al., 2021), which jointly pre-trains a visual and a textual encoder using an InfoNCE objective (Gutmann & Hyvärinen, 2010) with around 400 million curated image-text pairs. ALIGN (Jia et al., 2021) improves CLIP by scaling up the training dataset to 1.8 billion noisy image-text pairs, and BASIC (Pham et al., 2021) further increases the scale of both data and model. As a result, such CLIP-like models allow zero-shot inference when it comes to transfer learning on downstream tasks.

**Prompt tuning.** While fine-tuning a pre-trained model for downstream machine learning tasks has traditionally dominated the field of transfer learning, recent progress in prompt learning offers a compelling alternative. Specifically, the prompt tuning techniques fine-tune learnable prompt tokens attached to CLIP's inputs instead of training the entire model (Zhou et al., 2021; 2022; Wang et al., 2023; Yao et al., 2023). There also exist prompt tuning protocols for visual modality (Jia et al., 2022) and both visual and textual modalities (Yao et al., 2021; Zang et al., 2022). Similarly, there are adapter-based methods designed for CLIP-like models, which also freeze the encoders and only fine-tune several newly attached layers on top of them (Gao et al., 2021; Zhang et al., 2021).

## 3 Preliminaries

### 3.1 Contrastive Language-Image Pre-training

CLIP is a weakly supervised learning paradigm that combines visual and language encoders to solve image recognition problems. Formally, CLIP has an image encoder $\boldsymbol{F}_V : \mathbb{R}^{3 \times w \times h} \to \mathbb{R}^d$ where $w$ and $h$ denotes the input image's spatial resolution and $d$ denotes the dimension of the latent space, and a text encoder $\boldsymbol{F}_T : \mathbb{R}^{l \times d_e} \to \mathbb{R}^d$ where $l$ is the length of input sentence and $d_e$ is the dimension of word embedding. CLIP is trained by image-text pairs, in which the text briefly describes the information in the image. By encoding both image and text into the same latent space, CLIP can learn an alignment between visual and textual input with a contrastive loss (Gutmann & Hyvärinen, 2010). During inference, CLIP supports zero-shot classification by matching the visual representation of input image and the textual representation of target class names.

### 3.2 Prompt Tuning for Vision and Language

Despite CLIP's impressive zero-shot inference capabilities, there remains a noticeable accuracy gap in comparison to in-domain fine-tuning. However, fine-tuning the CLIP model may easily break the well-established alignment between vision and language, and CLIP will therefore lose the ability of open-vocabulary inference. Instead, prompt tuning attaches learnable tokens to the input, leaving the feature encoders fixed, which allows the model to retain its zero-shot and open-set inference abilities while significantly improving its in-domain accuracy.

**Textual Prompt Tuning (TPT).** As previously mentioned, CLIP's text query consists of a hand-crafted prompt (also referred to as prefix) such as "A photo of a" and a class name such as "dog". TPT replaces the prefix by learnable vectors (Zhou et al., 2021). During training, both CLIP's vision and language encoders are frozen and only the prompt vectors are optimized.

**Visual Prompt Tuning (VPT).** The prompt tuning protocol also works for visual input if the image encoder is a transformer-like model such as the Vision Transformer (Dosovitskiy et al., 2021). Specifically,

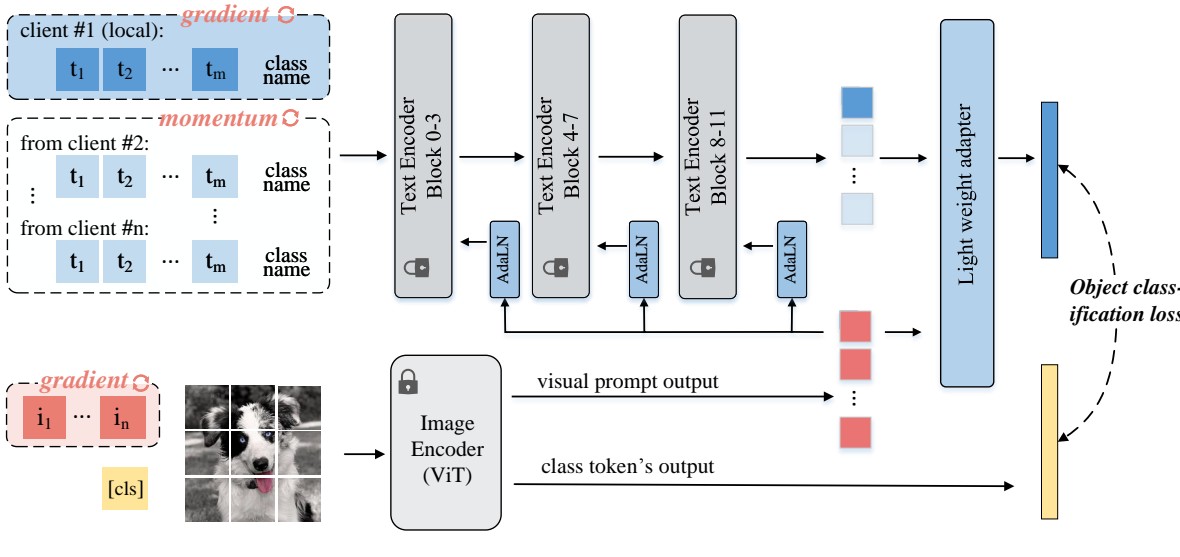

Figure 1: Local training framework. We load a pre-trained CLIP model and freeze both its image and text encoders. For each client, we feed the text encoder with n text prompts followed by class names, where one is optimized by the gradients and the rest $n-1$ are loaded from other clients with momentum update. We feed the image encoder with n learnable prompt tokens followed by patch-wise embedded images, where the prompt tokens are optimized by gradients.

this method attaches trainable vectors to the patch-wise embedded image, and uses an additional head to project the output. In VPT, only the prompt tokens and the head are optimized.

## 4 Methodology

**Problem Formulation.** Suppose there are $n$ clients that desire to deal with the same machine learning problem, e.g., image classification with the same target categories. The $n$ clients possess their own training data that originate from $n$ distinct domains. In other words, each client stands for a specific domain. We simulate this scenario using domain adaptation datasets like DomainNet (Peng et al., 2019), which encompass images from six different domains including clipart, information graph, painting, quickdraw, real-world images, and sketch. As the image features exhibit significant variations across different domains, it is indeed a challenging task for federated optimization. However, it is a realistic scenario because many times, the data heterogeneity between clients arises from differences in feature distributions rather than label distributions. Notably, our setting is compatible with the task that clients have non-i.i.d. labels. In our ablation study, we also further divide each domain into five splits with non-i.i.d. categories.

### 4.1 Local training

**Textual prompts.** With CLIP, a very simple way to deal with domain shift is to use domain-aware prompt contexts for text queries. For example, in DomianNet, when we use prefix "a painting of a" for the painting domain, and use "a sketch of a" for the sketch domain, the predictions can be more accurate and robust. This idea is also referred to as domain-specific prompts (Ge et al., 2022), while employing learnable text prompts can further improve the predictive performance. Inspired by this observation, we propose domain-specific prompts for CLIP's text encoder. Formally, we define a text prompt by a sequence of learnable tokens:

$$\boldsymbol{P}_T = [\boldsymbol{t}]_1[\boldsymbol{t}]_2 \ldots [\boldsymbol{t}]_m \in \mathbb{R}^{m \times d_e}, \tag{1}$$

where $m$ is the length of prompt and each token $[\boldsymbol{t}]_i \in \mathbb{R}^{d_e}$ has the same dimension as CLIP's word embedding.

Figure 1 illustrates ADAPT's local training framework and process. We initialize ADAPT by loading the same CLIP model for each client and freezing the parameters of both the image encoder $\boldsymbol{F}_V$ and the text encoder $\boldsymbol{F}_T$. For our task, we have $n$ text prompts $\boldsymbol{P}_T^1, \boldsymbol{P}_T^2, \ldots, \boldsymbol{P}_T^n$ corresponding to the $n$ domains. During local training, the $n$ text prompts are shared among the clients, yet the $i$-th prompt $\boldsymbol{P}_T^i$ can only be trained by the $i$-th client (we will detail this mechanism later). We separately feed the encoder $\boldsymbol{F}_T$ with all the $n$ text prompts followed by a class name, leading to $n$ representation vectors $\boldsymbol{f}_T^1, \boldsymbol{f}_T^2, \ldots, \boldsymbol{f}_T^n$, where

$$\boldsymbol{f}_T^i = \boldsymbol{F}_T(\boldsymbol{P}_T^i, [\text{class name}]). \tag{2}$$

Note that we assume each $\boldsymbol{f}_T^i$ stands for the representation of the class name in the $i$-th domain.

**Visual prompts.** We define visual prompts by $n$ learnable tokens $[\boldsymbol{v}]_1, [\boldsymbol{v}]_2, \ldots, [\boldsymbol{v}]_n$ which also correspond to the $n$ domains. During local training, we feed the visual encoder $\boldsymbol{F}_V$ (ViT architecture) with a class token [cls] (directly loaded from CLIP), $n$ visual prompts, and the patch-wise embedded image, leading to an image representation vector

$$\boldsymbol{f}_V = \boldsymbol{F}_V([\text{cls}], [\boldsymbol{v}]_1, [\boldsymbol{v}]_2, \ldots, [\boldsymbol{v}]_n, [\text{image}]). \tag{3}$$

We obtain the final textual representation through a cross attention layer. To minimize the number of parameters as much as possible, here we replace the query, key, and value projection matrices in the cross attention block with identity matrices, which we empirically find does not affect performance too much. Formally, denoting $\boldsymbol{q}_{\text{cls}}$ as the query vector of the class token, and $\boldsymbol{k}_i$ as the key vector of the $i$-th prompt token in $\boldsymbol{F}_V$'s last self-attention block, we have $\boldsymbol{w} = [w_1, w_2, \ldots, w_n]$ with

$$\boldsymbol{w}_i = \frac{\exp(<\boldsymbol{q}_{\text{cls}}, \boldsymbol{k}_i> /\tau_d)}{\sum_j \exp(<\boldsymbol{q}_{\text{cls}}, \boldsymbol{k}_j> /\tau_d)}, \tag{4}$$

where $\tau_d$ is a temperature coefficient. We regard each component $w_i$ as the visual feature's correlation to the $i$-th domain, and compute the final text output by

$$\boldsymbol{f}_T = \sum_{i=1}^n w_i \boldsymbol{f}_T^i. \tag{5}$$

During training, we optimize the model (actually learnable parameters only appear in prompts) using object classification loss, which is a cross-entropy function applied between $\boldsymbol{f}_V$ and $\boldsymbol{f}_T$, and a domain correspondence loss which is another cross entropy function applied between each pair of visual and textual outputs. Here we explain how we optimize these parameters. We desire the $i$-th text prompt $\boldsymbol{P}_T^i$ to represent the features of the $i$-th domain in the latent space of textual embeddings. However, the $i$-th client only possesses images from the $i$-th domain, so we cannot train $\boldsymbol{P}_T^j$ ($j \neq i$) yet instead load them from other clients. We introduce visual prompts to detect the correlations between an input image and the $n$ domains, so it is fine to optimize all of them. A detailed comparison of different training strategies can be found in our ablation study (see Table 5a and 5b for details).

**Guiding textual output by visual clues.** To facilitate domain-specific representations, we additionally introduce conditioning modules to guide textual outputs. The underlying idea of this strategy is that through the image encoder, we can effectively extract the domain information from the visual input; this domain information can help the text encoder to produce domain-specific outputs so that to accurately match the visual representation. In this work, we follow the prior practice of diffusion models (Peebles & Xie, 2023) to perform conditioning by AdaLN (Perez et al., 2018) which is efficient in parameter count. To process the textual features after the CLIP's text encoder, we employ a light-weight adapter which comprises a self-attention layer. We concatenate the text features with the output of visual prompts to further encourage domain-aware representations.

## 4.2 Parameters Aggregation

As mentioned above, for the $i$-th client, we optimize $\boldsymbol{P}_T^i$ by using gradients and load $\boldsymbol{P}_T^j$ ($j \neq i$) from other clients, so the aggregation of text prompts does not involve parameter merging processes (e.g. averaging).

| Method | #Com. cost | DomainNet | | | | | | |
|---|---|---|---|---|---|---|---|---|
| | | clipart | info g. | paint. | quick d. | real | sketch | avg. |
| Zero-Shot CLIP (Radford et al., 2021) | – | 66.1 | 40.6 | 62.3 | 13.5 | 80.4 | 58.5 | 53.6 |
| Single-Domain Tuning | – | 72.3 | 47.2 | 67.1 | 18.8 | 83.6 | 65.8 | 59.1 |
| *Conventional domain generalization methods:* | | | | | | | | |
| MIRO† (*ResNet-50*) (Cha et al., 2022) | 25.7M | 40.6 | 58.3 | 54.3 | 34.5 | 75.0 | 61.5 | 54.0 |
| PCL† (*ResNet-50*) (Yao et al., 2022b) | 25.6M | 41.2 | 59.9 | 54.8 | 32.9 | 74.1 | 61.5 | 54.1 |
| Fishr† (*ResNet-50*) (Rame et al., 2022) | 25.6M | 39.8 | 54.5 | 53.6 | 33.7 | 73.9 | 59.7 | 52.5 |
| *Conventional federated learning methods:* | | | | | | | | |
| FedAvg (*ResNet-50* ) (McMahan et al., 2017) | 25.6M | 40.2 | 61.1 | 57.6 | 33.5 | 75.6 | 60.3 | 54.7 |
| FedAvg (*ViT-B/16* ) (McMahan et al., 2017) | 85.8M | 42.4 | 60.7 | 57.0 | 30.4 | 79.8 | 61.1 | 55.2 |
| FedProx (*ResNet-50* ) (Li et al., 2020b) | 25.6M | 41.5 | 62.0 | 56.8 | 34.9 | 79.2 | 62.6 | 56.2 |
| FedProx (*ViT-B/16*) (Li et al., 2020b) | 85.8M | 40.5 | 63.1 | 57.4 | 29.7 | 81.2 | 59.8 | 55.3 |
| FedBN† (*ResNet-50*) (Li et al., 2021) | 45.4M | 35.7 | 43.6 | 41.0 | 30.1 | 67.7 | 44.6 | 43.8 |
| FPL† (*ResNet-50*) (Huang et al., 2023) | 25.6M | 40.5 | 49.2 | 56.9 | 35.0 | 70.2 | 60.7 | 52.1 |
| *Domain-agnostic vision-language tuning methods:* | | | | | | | | |
| PromptFL (Guo et al., 2023) | <5M | 76.0 | 50.2 | 70.4 | 33.5 | 81.2 | 67.8 | 63.2 |
| FedCLIP (Lu et al., 2023) | <5M | 74.1 | 48.3 | 68.5 | 31.8 | 80.5 | 58.6 | 60.3 |
| pFedPG (Yang et al., 2023) | <5M | 73.9 | 49.2 | 69.8 | 32.2 | 81.4 | 62.6 | 61.5 |
| FedOPT (Li et al., 2024) | <5M | 76.1 | 60.1 | 65.2 | 34.2 | 81.3 | 59.2 | 62.7 |
| FedAPT (Su et al., 2024) | <5M | 76.3 | 49.8 | 69.2 | 35.7 | 81.5 | 68.2 | 63.5 |
| FedTPG† (Qiu et al., 2024) | 21M | 75.4 | 50.3 | 69.3 | 33.5 | 81.0 | 67.4 | 62.8 |
| ADAPT (ours) | <5M | **78.5** | **64.7** | **72.5** | **44.5** | **85.8** | **73.2** | **69.8** |

Table 1: Test accuracy (%) on **DomainNet**. The *info g.*, *paint.*, and *quick d.* denote the domains of *infogragh*, *painting*, and *quickdraw*, respectively. Communication cost is measured by the number of learnable parameter shared at each round. † indicates reproduced results as experimental setup differs. Our results are marked in blue . The best results in each domain are **bolded**.

Suppose there is a centralized parameter server — although ADAPT also works for decentralized communication — and the clients upload their corresponding text prompt to it in each communication round. The server concatenates the $n$ uploaded text prompts and sends to every client. For visual parameters, as all visual prompts are optimized by every client, we perform federated averaging in the server and then send the merged parameters to each client. Note that we do not need to share the CLIP encoders' parameters as each client is initialized with the same CLIP model and its parameters are frozen during training.

This parameter aggregation paradigm works well for ADAPT, yet may create a minor problem for the text encoder. Specifically, after each communication round, the external text prompts of the $i$-th client, i.e., $\boldsymbol{P}_T^j$ ($j \neq i$) will be re-loaded. We observe that this sudden change of parameters often negatively affects our model. To address this issue, we propose to apply momentum update (also referred to as exponential moving average) to the external text prompts. Formally, we have

$$[\boldsymbol{t}]^s = \alpha [\boldsymbol{t}]^{s-1} + (1 - \alpha)[\boldsymbol{t}], \tag{6}$$

where $[\boldsymbol{t}]^s$, $[\boldsymbol{t}]^{s-1}$ denote the prompt tokens at the $s$ and $s-1$ step, and $[\boldsymbol{t}]$ denotes the vector received from other clients, and $\alpha \in [0, 1]$ is a coefficient to control the smoothness. The details of our ablation study related to momentum update are presented in Table 5a.

## 5 Experiments

**Datasets and Baselines**. We evaluate the proposed ADAPT and baseline methods on three domain adaptation image classification benchmarks: the DomainNet (Peng et al., 2019), OfficeHome (Venkateswara et al., 2017), and PACS (Li et al., 2017) datasets, presented in the Appendix. We first consider the baselines of CLIP and its adapted models to federated learning. The *Zero-shot CLIP*, which infers by aligning images to class names with a hand-crafted prompt, is a direct baseline to evaluate whether in-domain tuning is necessary for vision-language models in federated learning. We also introduce *Single-domain tuning*, which

applies textual prompt tuning (Zhou et al., 2021) to CLIP only in the local domain, as another baseline to testify whether it is helpful to combine the information across multiple domains. There are also domain-agnostic federated learning approaches based on CLIP such as *PromptFL* (Guo et al., 2023), *pFedPG* (Yang et al., 2023), *FedAPT* (Su et al., 2024), *FedOPT* (Li et al., 2024), *FedTPG* (Qiu et al., 2024) and *FedCLIP* (Lu et al., 2023), which train text prompt and an adapter layer in federated learning fashion, respectively. To further validate the effectiveness of ADAPT, comparisons were made with established federated learning algorithms such as *FedAvg* (McMahan et al., 2017) and *FedProx* (Li et al., 2020b), as well as recent approaches like *FedBN* (Li et al., 2021) and *FPL* (Huang et al., 2023) that address domain shift challenges not based on CLIP. Additionally, we compared traditional domain generalization methods including *MIRO* (Cha et al., 2022), *PCL* (Yao et al., 2022b), and *Fishr* (Rame et al., 2022), which also do not utilize CLIP. For these comparisons, we equipped the baselines with a 50-layer ResNet (He et al., 2016) and a base-scale Vision Transformer with a 16×16 patch size (Dosovitskiy et al., 2021), both pretrained on ImageNet-1k (Deng et al., 2009).

**Implementation details**. We employ a pre-trained CLIP model(Radford et al., 2021) with a ViT-Base/16 image encoder, so each textual and visual prompt token has the dimension of 512 and 768, respectively. We set the length of each textual prompt sequence $m = 16$ for better robustness, which follows the practice of TPT (Zhou et al., 2021). By default, the number of clients is determined by the number of domains for each dataset, i.e. $n = 6$ for DomainNet and $n = 4$ for OfficeHome and PACS. We train both our model and the baseline models for 200 epochs and execute the aggregation or broadcast process after every one epoch. We train the ResNet-based models and prompt tokens by a SGD optimizer with 0.01 learning rate, 0.9 momentum, and 0.005 weight decay. ADAPT instead uses AdamW (Loshchilov & Hutter, 2019) optimizer with $\beta_1 = 0.9$, $\beta_2 = 0.999$, 5e-4 learning rate, and 0.01 weight decay for transformer-based models. We set the temperature coefficient $\tau_d = 0.1$ in Equation 4, and set the momentum update ratio $\alpha = 0.99$ in Equation 6. If not specified, all reported results are average numbers over three trials.

## 5.1 Main Results

Table 1 shows that ADAPT significantly outperforms baseline methods on DomainNet, with notably high improvement in the "quickdraw" domain at 44.5% accuracy. This underscores the effectiveness of our prompt learning approach, which requires fewer trainable parameters, enhancing robustness even with larger models. In contrast, traditional methods like FedAvg and FedProx show minimal or negative gains, especially when upgrading from ResNet-50 to ViT-Base. ADAPT also achieves higher average accuracy and lower standard deviation compared to domain-agnostic methods, demonstrating better resilience against domain shifts. We further evaluate the models on OfficeHome and PACS datasets, and the results are summarized in Table 2. The experiments on these benchmarks also support our conclusion of ADAPT's effectiveness by demonstrating higher average accuracy and lower deviation across domains. Specifically, we improve the zero-shot CLIP by 16.2% average accuracy and 0.82% standard deviation over four domains in OfficeHome. We also observe that overall, the prompt-based methods consistently outperform the conventional federated learning algorithms that require to train the entire model. This confirms the benefits of employing parameter-efficient approaches in federated learning, and validates our choice of using prompt tuning to address the domain shift issues.

**Communication Costs.** ADAPT markedly reduces communication overhead in federated learning by only transferring domain prompts, contrary to standard methods that share all trainable parameters. To provide a clear comparison, we have included the following results in Table 1. An additional benefit of this approach is its ability to produce favorable results without requiring a substantial volume of training data. As shown in Table 8 (in the supplementary material), we obtain competitive few-shot results by our prompt tuning technique. In practice, We avoid fine-tuning the CLIP model to maintain its visual-language alignment. Fine-tuning large models such as CLIP escalates communication expenses and impedes the rate of convergence. With an equivalent number of training iterations, the fine-tuning protocol often falls short to prompt learning.

| Method | OfficeHome | | | | | PACS | | | | |
|---|---|---|---|---|---|---|---|---|---|---|
| | Ar | Cl | Pr | Rw | Avg. | P | A | C | S | Avg. |
| Zero-Shot CLIP | 79.5 | 63.1 | 85.3 | 86.5 | 78.6 | 99.8 | 96.9 | 98.8 | 87.7 | 95.8 |
| Single-Domain | 80.0 | 65.2 | 87.5 | 86.9 | 79.9 | 99.8 | 97.2 | 99.1 | 88.9 | 96.3 |
| *Conventional domain generalization methods:* | | | | | | | | | | |
| MIRO† (*ResNet-50*) | 64.2 | 48.9 | 77.3 | 72.3 | 65.7 | 90.5 | 53.3 | 78.3 | 77.2 | 74.8 |
| PCL† (*ResNet-50*) | 61.8 | 45.2 | 75.9 | 73.6 | 64.1 | 89.4 | 52.7 | 77.0 | 75.9 | 73.8 |
| *Conventional federated learning methods:* | | | | | | | | | | |
| FedAvg (*ResNet-50*) | 66.3 | 49.4 | 77.1 | 77.9 | 67.7 | 89.6 | 52.5 | 78.6 | 76.1 | 74.2 |
| FedAvg (*ViT-B/16*) | 67.9 | 49.6 | 77.5 | 81.0 | 69.0 | 91.3 | 54.8 | 79.2 | 77.9 | 75.8 |
| FedProx (*ResNet-50*) | 68.8 | 50.5 | 78.6 | 80.3 | 69.6 | 91.7 | 57.0 | 81.8 | 80.2 | 77.7 |
| FedProx (*ViT-B/16*) | 70.4 | 51.3 | 80.3 | 82.4 | 71.1 | 92.0 | 59.4 | 83.5 | 81.6 | 79.1 |
| FedBN (*ResNet-50*) | 71.2 | 51.6 | 81.5 | 83.6 | 72.0 | 92.5 | 63.8 | 85.2 | 84.3 | 81.5 |
| FPL (Huang et al., 2023) | 63.7 | 54.6 | 76.5 | 75.4 | 67.6 | 90.2 | 51.3 | 86.4 | 84.7 | 78.2 |
| GA (Zhang et al., 2023) | 67.4 | 52.9 | 78.3 | 78.5 | 69.3 | 91.1 | 52.4 | 99.1 | 78.2 | 80.2 |
| *Domain-agnostic vision-language tuning methods:* | | | | | | | | | | |
| PromptFL (Guo et al., 2023) | 79.8 | 65.6 | 89.5 | 89.1 | 81.0 | **99.9** | 97.1 | 99.0 | 90.6 | 96.7 |
| FedCLIP (Lu et al., 2023) | 79.1 | 65.0 | 88.6 | 88.4 | 80.3 | 99.8 | 97.4 | 98.9 | 89.0 | 96.3 |
| Ours | **83.1** | **69.6** | **90.5** | **90.4** | **83.4** | **99.9** | **98.3** | **99.2** | **91.7** | **97.3** |

Table 2: Test accuracy (%) on **OfficeHome** and **PACS**. Domains include *art*, *clipart*, *product*, and *real-world* for OfficeHome, and *photo*, *art painting*, *cartoon*, and *sketch* for PACS. Our results are marked in blue . The best results are **bolded**.

| Method | Fed. | VPT | TPT | domain | AdaLN | acc. |
|---|---|---|---|---|---|---|
| Zero-Shot CLIP | ✗ | ✗ | ✗ | ✗ | ✗ | 53.6 |
| Single-Domain | ✗ | ✗ | ✓ | ✗ | ✗ | 59.1 |
| Visual Only | ✓ | ✓ | ✗ | ✗ | ✗ | 54.2 |
| Textual Only | ✓ | ✗ | ✓ | ✗ | ✗ | 63.2 |
| Domain-Agnostic | ✓ | ✓ | ✓ | ✗ | ✗ | 63.5 |
| Prompt-only | ✓ | ✓ | ✓ | ✓ | ✗ | 68.4 |
| ADAPT | ✓ | ✓ | ✓ | ✓ | ✓ | 69.8 |

Table 3: Ablation study to model components. We report the average accuracy (%) over six domains in DomainNet. *VPT* and *TPT* denote whether using visual or textual prompts.

## 5.2 Ablation Studies

We first dissect the ADAPT model to ablate its performance gains. ADAPT comprises two primary components: visual prompts and domain-specific text prompts. By dissecting these components, we get three more variants of our method: 1) *Visual Only*, which leverages learnable prompt tokens for only image input and uses CLIP's hand-crafted prompt for texts. 2) *Textual Only*, which discards the visual prompt tokens of ADAPT and uses learnable text prompts only. Note that in the absence of visual prompts, we cannot get the weight $w_i$ (see Equations 4 and 5) for each domain, so the text prompts from external clients should also be discarded. We instead aggregate the textual prompts by federated averaging (McMahan et al., 2017). 3) *Domain-Agnostic*, which retains both ADAPT's visual and textual prompts but decouples them, i.e., we do not perform the weighted sum process in Equation 5, which can be considered as a simple combination of the modes *Textual Only* and *Visual Only*.

We summarize the results in Table 3. Since we introduce visual prompt tuning to combine domain information rather than enhancing the visual feature extraction abilities, we do not attach an additional head for the image encoder as in (Jia et al., 2022). Therefore, the *Visual Only* mode cannot yield significant performance

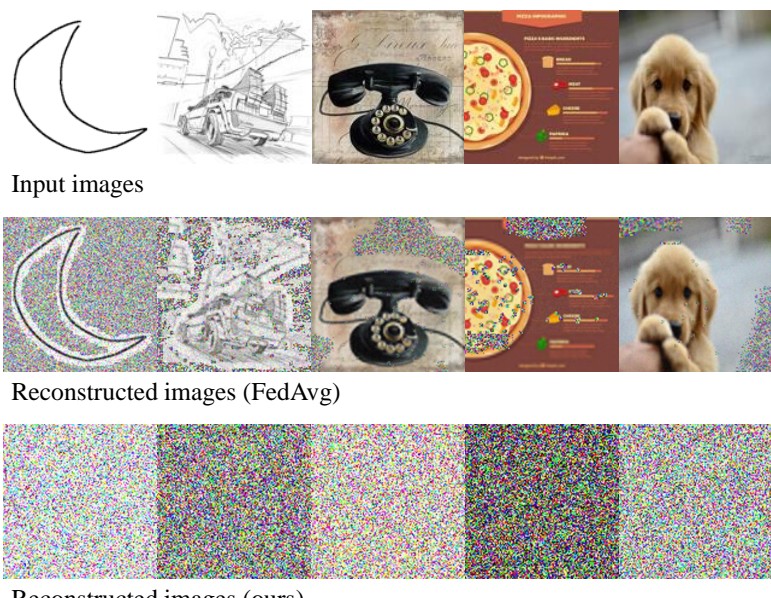

Input images

Reconstructed images (FedAvg)

Reconstructed images (ours)

Figure 2: Examples of reconstructed images produced by *Deep Leakage from Gradient* (Zhu et al., 2019). It shows that the gradient attack algorithm cannot reconstruct semantically meaningful information from ADAPT as we only share few parameters.

improvements. We also observe that tuning textual prompts results in a 5.5% increase in accuracy, and when tuning them in a federated learning fashion, we achieve an additional 4.1% improvement (*Textual Only*). Notably, compared to the simple visual-and-textual prompt tuning with 63.5% accuracy, ADAPT achieves a much higher result of 69.8%, demonstrating the crucial significance of our domain-aware design.

**Momentum update, prompt length, and communication frequency.** We consider three more factors that may affect results. As mentioned in Section 4, we update the external text prompts by exponential moving average to prevent parameters' sudden change. Table 5a presents comparisons regarding the update mechanism for text prompts, where the accuracy drops by 2.6% in the absence of momentum update. If we train all text prompt tokens in every client, i.e., we disregard the relationship between text prompts and domains, the accuracy drops by 4.7% as it makes ADAPT a domain-agnostic approach.

By default, we aggregate the visual prompt tokens by federated averaging, as separately training each token in a specific domain does not yield better performance (see Table 5b). As shown in Table 5e, we set each textual prompt length to $m = 16$, as it works more robust than a shorter prompt ($m = 4$), and when we further increase the length, the model tends to overfit and accuracy drops. Notably, the visual and textual prompt lengths are consistent. In Table 5d we also assess the impact of communication frequency by varying it to 0.5, 1, and 2 training epochs per communication round. It shows that compared to our default setup of one epoch per communication round, more frequent aggregation (0.5 epoch/round) does not lead to improved performance, while conversely, infrequent communication (2 epochs/round) results in a 0.7% accuracy degradation.

**AdaLN component.** We evaluated the effect of varying the number of AdaLN layers in our model. Beyond the default three layers, we tested configurations with an AdaLN layer after every two blocks in the vision encoder (six layers total) and after each block (12 layers total). As shown in Figure 5f, increasing from three to 12 AdaLN layers resulted in only a 0.2% accuracy improvement, which is minimal considering the additional network parameters. Thus, we retained the default three-layer configuration. Replacing the AdaLN layers with cross-attention mechanisms improved performance by 0.7% compared to the model without AdaLN. Using AdaLN layers provided an additional 0.7% improvement over the cross-attention approach, as illustrated in Figure 5c.

| # | clipart | infograph | painting | quickdraw | real | sketch |
|---|---------|-----------|----------|-----------|------|--------|
| 1 | ~ | fe | N/A | N/A | ° | kd |
| 2 | N/A | # | dng | , | ... | with |
| 3 | lh | bh | some | ? | N/A | N/A |
| 4 | and | N/A | lh | N/A | the | pjf |

Table 4: Nearest Words of textual prompts learned by ADAPT in DomainNet dataset. N/A means non-Latin characters. It shows that our prompts tend to capture high-level and abstract semantics that are difficult to summarize using standard natural language words found in the dictionary.

| mode | acc. |
|------|------|
| w/ mtm. | **69.8** |
| w/o mtm. | 67.2 |
| train all | 65.1 |

(a) Text prompt update.

| mode | acc. |
|------|------|
| average | **69.8** |
| split w/ mtm. | 69.6 |
| split w/o mtm. | 68.5 |

(b) Visual prompt update.

| mode | acc. |
|------|------|
| w/ AlaLN | 69.8 |
| w/o AlaLN | **68.4** |
| cross attention | 69.1 |

(c) Visual conditioning.

| #eps/round | acc. |
|------------|------|
| 0.5 | **69.8** |
| 1 | **69.8** |
| 2 | 69.1 |
| 3 | 68.4 |

(d) Comm. frequency

| #tokens | acc. |
|---------|------|
| 4 | 67.8 |
| 9 | 68.5 |
| 16 | **69.8** |
| 32 | 68.3 |

(e) Prompt length.

| numbers | acc. |
|---------|------|
| 0 | 68.4 |
| 3 | 69.8 |
| 6 | 69.8 |
| 12 | **70.0** |

(f) AlaLN amount.

| learn. rate | acc. |
|-------------|------|
| $5 \times 10^{-5}$ | 66.9 |
| $1 \times 10^{-4}$ | 67.2 |
| $5 \times 10^{-4}$ | **69.8** |
| $1 \times 10^{-3}$ | 69.1 |
| $5 \times 10^{-3}$ | 67.4 |

(g) Learning rate.

| weight decay | acc. |
|--------------|------|
| 0.001 | 69.0 |
| 0.005 | 69.5 |
| 0.01 | **69.8** |
| 0.05 | 69.4 |
| 0.1 | 68.8 |

(h) Weight decay.

| epochs | acc. |
|--------|------|
| 100 | 64.1 |
| 200 | **69.8** |
| 300 | 70.2 |
| 400 | 70.4 |
| 600 | 70.3 |

(i) Training epochs.

Table 5: Ablation studies. We report the average accuracy over six domains in DomainNet. The *mtm.* denotes momentum update. Our default setup is marked in blue. The best results of each ablation study is **bolded.**

**Privacy Preservation.** In ADAPT, there are two potential ways to expose participants' private data. First, similar to most federated learning algorithms, ADAPT shares gradient information across all participants, so some private information might be able to be reconstructed by gradient-based attacking algorithms such as *Deep Leakage from Gradient* (DLG) (Zhu et al., 2019). However, as ADAPT has relatively small number of learnable parameters, such attacks cannot extract sufficient information from gradients to reconstruct participants' local data, which gives ADAPT a significant privacy advantage over traditional federated learning algorithms. Figure 2 presents examples of DLG applied to our model and FedAvg. While ADAPT shows no significant information leakage, attacks on FedAvg reveal extensive details from the original image.

Another potential way to expose privacy is decoding the trained text prompts, which might contain some statistical information of participants. However, our experiments showcase that this is difficult as well. we follow CoOp (Zhou et al., 2021) to decode each text prompt by finding a standard vocabulary word with minimum Euclidean distance to it in the embedding space, and summarize the interpretation results for DomainNet in Table 4. It shows that our prompts tend to capture some high-level and abstract semantics that are difficult to be summarized to standard natural words.

# 6  Conclusion

This work introduces ADAPT, a novel federated learning approach explicitly designed to address the key challenges of domain shift and communication efficiency. Our method strategically combines CLIP and prompt learning techniques for both visual and textual inputs, thereby enhancing parameter-efficiency and minimizing communication costs, while maintaining robustness in federated optimization involving heterogeneous data. Furthermore, we address the pervasive issue of domain shift across clients by introducing domain-specific prompts and facilitating correlations between visual and textual representations through self-attention mechanisms. These innovations result in a domain-aware federated learning methodology that consistently demonstrates outstanding effectiveness. Notably, our experiments reveal a remarkable achievement—an average accuracy of 69.8% across six domains in the DomainNet dataset, marking an impressive 16.2% improvement over the original CLIP model. In comparisons with traditional federated learning methods like FedAvg and FedProx, as well as existing domain-agnostic CLIP-based approaches such as PromptFL and FedCLIP, our ADAPT consistently outperforms them across three benchmark scenarios.

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
