# A    Appendix

| Method | DomainNet | | | | | | |
|---|---|---|---|---|---|---|---|
| | clipart | infograph | painting | quickdraw | real | sketch | average |
| Zero-Shot CLIP | 66.1 | 40.6 | 62.3 | 13.5 | 80.4 | 58.5 | 53.6 |
| FedAvg | 37.6 | 56.4 | 55.6 | 31.0 | 71.9 | 57.2 | 51.6 |
| FedProx | 38.4 | 57.2 | 54.9 | 32.5 | 72.8 | 58.5 | 52.4 |
| PromptFL | 73.2 | 48.1 | 68.7 | 31.9 | 78.6 | 64.7 | 60.9 |
| FedCLIP | 72.7 | 47.0 | 66.2 | 32.8 | 76.9 | 57.2 | 58.8 |
| ADAPT (ours) | **75.9** | **63.3** | **72.3** | **40.9** | **84.2** | **72.4** | **68.2** |

Table 6: Test accuracy (%) on DomainNet with 30 clients. Our results are marked in blue . The best results in each domain are **bolded**.

| Method | CLIP-based | full | 1-shot | 2-shot | 4-shot | 8-shot | 16-shot |
|---|---|---|---|---|---|---|---|
| Single Domain Tuning | ✓ | 59.1 | 51.1 | 51.8 | 53.2 | 54.7 | 56.2 |
| FedAvg (*ResNet-50*) | ✗ | 54.7 | - | - | - | - | 15.1 |
| FedAvg (*ViT-Base/16*) | ✗ | 55.2 | - | - | - | - | 19.7 |
| PromptFL | ✓ | 63.2 | 51.4 | 51.8 | 55.2 | 57.6 | 61.2 |
| FedCLIP | ✓ | 60.3 | 50.8 | 51.2 | 52.1 | 53.4 | 54.6 |
| ADAPT (ours) | ✓ | **69.8** | **55.2** | **57.4** | **61.4** | **64.4** | **65.7** |

Table 7: Few-shot accuracy (%) on DomainNet. $n$-shot denotes training with $n$ samples per class and per domain. Our results are marked in blue . The best results are **bolded**.

**Datasets.** We evaluate our ADAPT and baseline methods on the following three domain adaptation image classification benchmarks:

- DomainNet Peng et al. (2019). The DomainNet dataset has around 600,000 images spanning 345 categories from six domains, which covers diverse image styles including clipart, infograph, painting, quickdraw, real, and sketch.

- OfficeHome Venkateswara et al. (2017). The OfficeHome dataset consists of approximately 15,500 images depicting everyday objects in 65 classes. It further categorizes the images into four domains: art, clipart, product, and real-world.

- PACS Li et al. (2017). The PACS dataset contains around 10,000 images drawn from seven categories and four domains, including photo, sketch, cartoon, and painting styles.

**Decentralization.** By default, we consider each domain in the dataset as a single client, leading to non-identical feature distributions yet the same class distribution across clients. To further validate our method's effectiveness and flexibility, we consider a more challenging scenario on DomainNet where each domain is further divided into five clients by Dirichlet sampling, leading to 30 sub-datasets with either non-i.i.d. features or non-i.i.d. categories. Under this setup, we average the text prompt tokens for clients in the same domain at the aggregation step. The results are summarized in Table 6. Compared to our default setting which each domain is considered as one client, ADAPT only has 1.6% accuracy decrease when the dataset is further divided. In contrast, the conventional methods FedAvg and FedProx perform more sensitive to the non-i.i.d categories, with 3.6% and 2.9% accuracy decrease, respectively.

**Robustness to few-shot learning.** One of the advantages of prompt learning is the robustness to few-shot scenarios. We investigate if our dual prompt tuning method retains this merit in the context of federated

learning. Therefore, we conduct few-shot learning experiments on DomainNet, employing 1, 2, 4, 8, and 16 training samples per category and per domain. We evaluate the other CLIP-based methods with the same setting, yet only test 16-shot performance for FedAvg as it fails to yield reasonable results with fewer training samples. The corresponding results are summarized in Table 7. As is shown, CLIP-based methods exhibit superior robustness against few-shot learning than FedAvg, which again demonstrates the significant benefits of using parameter-efficient approaches. Also, our ADAPT consistently outperforms the baselines in few-shot learning.

**Efficiency.** Below we provide a detailed comparison of training and inference throughput (images/second), and the number of learnable parameters for each client. The speed is tested on a single NVIDIA A100 GPU. As shown, the four listed methods share a similar level of training and inference speed, but our model has a clear advantage in performance.

| method | train (img/s) | infer (img/s) | #params (M) | acc. (%) |
|--------|---------------|---------------|-------------|----------|
| FedAvg | 175 | 676 | 85.8 | 55.2 |
| PromptFL | 81 | 624 | 0.08 | 63.2 |
| FedCLIP | 126 | 572 | 0.53 | 60.3 |
| Ours | 80 | 578 | 0.35 | **69.8** |

Table 8: Comparison of training and inference throughput (images per second), the number of learnable parameters, and accuracy for each client on a single NVIDIA A100 GPU. Our model achieves comparable efficiency while significantly improving performance.

---

**Algorithm 1** Training Process of ADAPT

---

**Input:**
    CLIP vision encoder $\boldsymbol{F}_V$, text encoder $\boldsymbol{F}_T$
    $n$ local datasets, each $\boldsymbol{D}_i = \{([\text{image}], [\text{class name}])_j\}_{j=1}^J$
    Total communication rounds $T$, momentum coefficient $\alpha$
**Initialization:**
    Randomly initialize text prompts $[\boldsymbol{P}_T^1]^0, \ldots, [\boldsymbol{P}_T^n]^0$
    Randomly initialize visual prompts $[\boldsymbol{V}] = \{[\boldsymbol{v}]_1, \ldots, [\boldsymbol{v}]_n\}$
    Broadcast the pretrained model and prompts to $n$ clients
1: **for** $t = 1$ to $T$ **do**
2:    *# Local training in parallel*
3:    **for** $i = 1$ to $n$ **do**
4:        Keep $\boldsymbol{F}_V$ and $\boldsymbol{F}_T$ frozen
5:        **for** $j = 1$ to $J$ **do**
6:            Compute $\boldsymbol{f}_T^k = \boldsymbol{F}_T(\boldsymbol{P}_T^k, [\text{class name}]_j)$ for $k \in \{1, \ldots, n\}$
7:            Compute $\boldsymbol{f}_V = \boldsymbol{F}_V([\text{cls}], [\boldsymbol{v}]_1, \ldots, [\boldsymbol{v}]_n, [\text{image}]_j)$
8:            Extract attention scores $\boldsymbol{w} = [w_1, \ldots, w_n]$ from $\boldsymbol{F}_V$ using Eq.5
9:            Weighted sum: $\boldsymbol{f}_T = \sum_{k=1}^n w_k \boldsymbol{f}_T^k$
10:          Compute L$_2$ loss: $\mathcal{L} = <\boldsymbol{f}_V, \boldsymbol{f}_T> / ||\boldsymbol{f}_V|| \cdot ||\boldsymbol{f}_T||$
11:          Update $[\boldsymbol{v}]_1, \ldots, [\boldsymbol{v}]_n$ and $\boldsymbol{P}_T^i$ by $\mathcal{L}$
12:          Update $\boldsymbol{P}_T^k, k \in \{1, \ldots, n\}, k \neq i$ by momentum: $\boldsymbol{P}_T^k = \alpha \boldsymbol{P}_T^k + (1 - \alpha)[\boldsymbol{P}_T^k]^{t-1}$
13:        **end for**
14:    **end for**
15:    *# Global aggregation in the server*
16:    Average $[\boldsymbol{V}] = \frac{1}{n} \sum_{k=1}^n [\boldsymbol{V}]^k$, where $[\boldsymbol{V}]^k = \{[\boldsymbol{v}]_1, \ldots, [\boldsymbol{v}]_n\}$ obtained from #$k$ client
17:    Assign $[\boldsymbol{P}_T^k]^t = \boldsymbol{P}_T^k$, where $\boldsymbol{P}_T^k$ obtained from #$k$ client
18:    Broadcast $[\boldsymbol{V}], [\boldsymbol{P}_T^k]^t (k \in \{1, \ldots, n\})$ to all clients
19: **end for**

---