# OpenReview forum: "Learning to Prompt Your Domain for Federated Vision-Language Models"
_TMLR — Accepted by TMLR_

### Review · Reviewer_3GXs · 2025-09-03

**Summary Of Contributions:**

This paper proposes ADAPT (Federated Domain-Aware Prompt Tuning), a method that applies domain-specific prompt learning to CLIP models in federated learning scenarios. The approach addresses domain shift issues by using separate visual and textual prompts for each domain, with a cross-attention mechanism to match images to appropriate domain-specific text representations.

**strengths**

1. addresses realistic federated learning scenario with domain heterogeneity
2. only 0.08M trainable parameters
3. Consistent improvements across three datasets (DomainNet, OfficeHome, PACS)
4. Comprehensive ablation studies and privacy analysis

**weaknesses**

1. only combines existing techniques (CLIP, prompt tuning, federated learning)
2. assumes one client per domain, limiting real-world applicability
3. Incomplete comparison with recent federated domain adaptation methods

**Audience:**

Yes

**Audience Explanation:**

The federated learning community would find this work relevant, particularly for applications with domain heterogeneity. The parameter efficiency results are practically important for communication-constrained scenarios. The combination of vision-language models with federated learning addresses a timely intersection of active research areas.

**Broader Impact Concerns:**

No significant ethical concerns

**Claims And Evidence:**

Yes

**Claims Explanation:**

The experimental validation contains proper baselines, ablation studies, and consistent results across datasets, e.g. the 14.8% improvement over zero-shot CLIP on DomainNet. Privacy claims are supported by gradient reconstruction experiments. However, the domain-client correspondence assumption limits the generalizability of findings.

**Requested Changes:**

**critical**

Expand evaluation to scenarios where multiple clients share domains (not one-to-one mapping)

Compare against recent federated domain adaptation methods beyond basic FedAvg/FedProx

Provide theoretical analysis or bounds on the approach's convergence properties

**non-critical**

Discuss computational overhead during inference (cross-attention mechanism)

Evaluate on larger-scale datasets beyond the current benchmarks

Add analysis of failure cases or method limitations

Include comparison with domain-agnostic prompt learning in centralized setting

---

> ### Author Response · Authors · 2025-09-20
> **Author Response to Reviewer 3GXs**
>
> We sincerely appreciate your thoughtful comments and constructive feedback. Below is detailed response.
>
> **Q1:** Multiple clients sharing the same domain.
>
> **A1:** Thanks for your feedback. We have already conducted these experiments, but the results are currently included in the Appendix. In the revised version, we will move them, along with the corresponding analysis, to the main text. Specifically, as shown in Table 6, we set up 30 clients across six domains in DomainNet, with every five clients assigned to a distinct domain. Under this setup, our method still consistently achieves the best performance compared to all baselines.
>
> **Q2:** More comparisons to federated domain adaptation methods.
>
> **A2:** Thank you for your advice. In Table 1, we compared our approach with FPL (Huang et al., 2023), a federated learning algorithm designed for domain shift challenges, where our ADAPT method significantly outperforms it in terms of both accuracy and communication efficiency on DomainNet. To further address your concern, we additionally compare against another federated domain generalization approach (denoted as FedAvg+GA [1]) and report results on the OfficeHome and PACS datasets. As shown below, these domain adaptation–focused baselines do not achieve results as competitive as ours, which we believe largely reflects the optimization advantages brought by the parameter-efficient nature of prompt learning. We will add these comparisons in the revised version.
>
> | Method|||OfficeHome|||||PACS|||
> |-----------------|:------:|:------:|:------:|:------:|:------:|:-----------------------:|:------:|:------:|:------:|:------:|
> || Ar | Cl | Pr| Rw | Avg. | P | A| C| S | Avg.|
> | FedAvg| 66.3 | 49.4 | 77.1 | 77.9 | 67.7 | 89.6 | 52.5 | 78.6 | 76.1 | 74.2 |
> | FPL| 63.7 | 54.6 | 76.5 | 75.4 | 67.6 | 90.2 | 51.3 | 86.4 | 84.7 | 78.2 |
> | FedAvg+GA| 67.4 | 52.9 | 78.3 | 78.5 | 69.3 | 91.1 | 52.4 | 99.1 | 78.2 | 80.2 |
> |**Ours** | **83.1** | **69.6** | **90.5** | **90.4** | **83.4** | **99.9** | **98.3** | **99.2** | **91.7** | **97.3** |
>
> [1] Zhang, Ruipeng, et al. "Federated domain generalization with generalization adjustment." CVPR 2023.
>
> **Q3:** Theoretical analysis or bounds on the approach's convergence properties.
>
> **A3:** Thank you for your suggestion. Under standard assumptions (detailed below) such as L-smoothness and bounded variance , ADAPT achieves the same order of convergence rate as FedAvg. The only additional factor is a controllable bias term introduced by EMA, whose coefficient is proportional to 1−α (where α is the EMA parameter, set to α=0.99 in our paper).
>
> Here we provide the error bound under the $\mu$-strongly convex condition for $F$; a more general proof under broader conditions will be included in the revision. Under assumptions of (1) The global objective F(θ) has L-Lipschitz continuous gradients, i.e., $L$-smooth; (2) Local stochastic gradients are unbiased estimators of the true gradient, and their variance is bounded by $\sigma^2$, i.e., bounded variance; (3) The deviation between each client’s gradient and the global gradient is bounded $\mathbb{E}\|\nabla f_i(\theta)-\nabla F(\theta)\|^2 \le \delta^2$, i.e., bounded data heterogeneity; (4) The squared norm of local gradients has a bounded expectation $\mathbb{E}\|\nabla f_i(\theta)\|^2 \le G^2$, i.e., bounded second moment of gradients; (5) a learning rate condition $\eta \le \tfrac{1}{2L}$, we have $\mathbb{E}\left[F(\bar\theta_R)-F^\star\right]\le
> O\left(\frac{1}{\eta R E}\right)+O\left(\eta\frac{\sigma^2}{K}\right)+O\left(\eta^2 E^2\delta^2\right)+O\left(\eta^2 E^2(1-\alpha)^2G^2\right)$, where $R$ denotes the total number of communication rounds and $E$ denotes the number of local gradient decent steps per round. The first three terms of this bound are identical to those of FedAvg, while the last term involving $E$ has the same order as the third term and can be controlled by $\alpha$.
>
> We will provide the complete proof in the revised version.
>
> **Q4:** Clarification of novelty and contributions.
>
> **A4:** Thanks for your insightful review. We agree that our method involves several concepts, including CLIP, prompt learning, and federated learning, but we would like to emphasize that this does not mean our approach is a simple combination of existing methods. Our motivation is to address the real-world challenge in federated learning where severe domain shifts across clients hinder model convergence. In this scenario, employing a pretrained model that encapsulates knowledge from multiple domains is crucial, and using CLIP is a common practice to tackle such issues (e.g., our baselines FedCLIP, FedOPT, and pFedPG). Within this context, prompt learning serves as our key strategy. Through a series of parameter-efficient and domain-aware designs, we significantly improve the predictive performance of federated learning models under domain-heterogeneous settings, offering valuable insights for real-world applications of federated learning.

---

> > ### Author Response · Authors · 2025-09-20
> > **For the non-critical questions**
> >
> > **Q1:** Computational overhead during inference.
> >
> > **A1:** Thank you for the question. Our components slightly lower the inference speed but attains significantly better predictive performance. For example, compared to a standard prompt learning framework like PromptFL, our inference throughput decreases from 624 to 578 images/second, but the accuracy improves from 63.2% to 69.8%, which we believe is a good trade-off between efficiency and accuracy. Compared to the standard FedAvg model, the throughput is 676 vs. 578, but the accuracy is substantially different --- the FedAvg model only achieves an accuracy of 55.2%. We will discuss it in the revised version.
> >
> > **Q2:** Evaluate on larger-scale datasets beyond the current benchmarks.
> >
> > **A2:** Thank you for your suggestion. We do consider scaling up the dataset as an important direction for future work. At present, DomainNet is one of the largest well-curated public dataset with domain annotations that we can access, and our baseline methods also primarily focus on data at this scale. We apologize for not being able to provide scaling-up results within the two-week rebuttal period, as this would require extensive data curation and much longer training time. However, we believe this is a very promising follow-up direction, and we will further discuss the scalability of ADAPT in future work.
> >
> > **Q3:** Add analysis of failure cases or method limitations.
> >
> > **A3:** Thanks for your advice. We acknowledge that one major challenge of our method is the need to set prompts for every domain. While this works well in general scenarios without introducing significant optimization or computational overhead, under extreme conditions where the number of domains is very large (e.g., >100), our method may require certain modifications to ensure convergence and maintain efficiency. One possible solution is to introduce a hierarchical design for domains and optimize prompts in a tiered manner. We will discuss these limitations in the revised version.
> >
> > **Q4:** Include comparison with domain-agnostic prompt learning in centralized setting.
> >
> > **A4:** Thank you for your insightful suggestion. Below we present a comparison between our method and centralized training. Both methods use a ViT-Base/16 as image encoder. The results suggest that our approach attains an accuracy that is close to its theoretical upper bound. We will add this result in the revision.
> >
> > | Method        | clipart | infograph | painting | quickdraw |  real | sketch |  avg. |
> > |---------------|--------:|----------:|---------:|----------:|------:|-------:|------:|
> > | Centralized |   80.4  |     75.1  |     76.0 |     59.4  |  87.2 |   76.5 |  75.8 |
> > | **Ours**      |   78.5  |     64.7  |     72.5 |     44.5  |  85.8 |   73.2 |  69.8 |

---

### Review · Reviewer_RceZ · 2025-09-11

**Summary Of Contributions:**

In this paper, the authors propose a CLIP-based federated prompt-tuning technique called ADAPT for vision-language models.

They introduce domain-specific tuning of textual prompts, making their approach domain-aware. In addition to learnable textual prompts, they also allow for visual prompts to be learned and combine the representations using attention mechanisms. Since it is a prompt-tuning technique, the authors state that it has the advantage of parameter efficiency and lower communication cost compared to other fine-tuning techniques.

In their experiments, they empirically demonstrate improved performance over federated baselines and other domain-agnostic tuning methods on three different datasets. Furthermore, they provide a comprehensive ablation study of the different components of their technique. Finally, they provide empirical evidence suggesting that clients private data cannot be easily recovered when using their technique in this federated setting.


### Strengths:

S.1. Proposing an efficient domain-aware tuning method for federated learning is an interesting direction.

S.2. The empirical results are comprehensive and show improved performance when using the proposed technique. The authors compare their technique with various baselines and other approaches across $3$ different domains.

S.3. The authors include an extensive ablation study, showcasing the importance of its component and different design choices with respect to final performance.


### Weaknesses:

W.1. Given the various CLIP-based federated techniques already presented in related work, I would have expected a more direct comparison highlighting how the proposed method differs from them. This comparison will help clarify the contributions of this paper.

W.2. My main concern is the scalability of the technique to a larger number of clients in federated settings. In the experiments, the authors assume one client per domain, which allows for only $6$ clients in the DomainNet dataset. However, in real-world federated scenarios, the number of clients is typically much larger. It would be helpful if the authors could discuss that.

W.3. I believe the claim of reduced communication cost is slightly overemphasized, since a similar communication cost is shared across other CLIP-based tuning methods.

W.4. It is not very clear to me whether prompt-tuning provides any advantage over other parameter-efficient tuning techniques, such as LoRA. For example, would ADAPT be comparable to LoRA-based finetuning in a federated setting?  [1]

W.5. The paper lacks a discussion of potential disadvantages of domain-specific tuning in federated learning. It would be beneficial to add such a discussion.


References

[1] Kuo et al., Federated LoRA with Sparse Communication, 2024.

**Audience:**

Yes

**Audience Explanation:**

I believe that proposing an efficient domain-aware tuning method for federated learning is an interesting direction to certain individuals in TMLR's audience. However, it would be helpful if the authors could address any weaknesses listed above.

**Claims And Evidence:**

Yes

**Claims Explanation:**

Most of the claims the authors make are supported by the empirical evidence. The results in the experiments and ablation section are convincing. See Strengths above.

See W.3 in the Weaknesses list.

W.3. I believe the claim of reduced communication cost is slightly overemphasized, since a similar communication cost is shared across other CLIP-based tuning methods.

**Requested Changes:**

I would like to ask the authors to address and clarify the aforementioned weaknesses (W1-W5). I believe addressing those will strengthen the paper.

Moreover, I have a few additional minor questions and suggestions:

- Regarding the implementation, why are different optimizers used? SGD optimizer for ResNet-based models and AdamW optimizer for the proposed technique.
- In Section 5.1, the authors mention that ADAPT achieves lower standard deviation compared to other methods, demonstrating better resilience in domain shifts. However, the standard deviation across domains is not reported.
- As a formatting recommendation, I suggest reordering Table 5 with Figure 2 and Table 4. Based on paper’s structure, it makes more sense to move Table 5 to page 9 and Figure 2 as well as Table 4 to page 10.

---

> ### Author Response · Authors · 2025-09-20
> **Author Response to Reviewer RceZ**
>
> We are grateful to your careful evaluation and insightful suggestions. Below is the detailed response to your questions and concerns.
>
> **Q1:** Methodology comparison to existing approaches.
>
> **A1:** Thank you for your constructive suggestion. Below is a detailed comparison between our ADAPT and the existing baselines. We will include this comparison in the revised version.
>
> | Method | Mechanism| Domain-Aware| VL Coupling | Aggregation protocol   |
> |-|-|-|-|-|
> | FedAvg | Full-model averaging | No  | N/A  | Average  |
> | ADAPT | Domain-specific text & visual prompts; image-conditioned AdaLN  | Yes | Dual-modal prompt and AdaLN  | Average and EMA  |
> | PromptFL | Soft text prompt  | No | CLIP similarity| Average |
> | FedCLIP  | Attention-based image adapter| No   | CLIP similarity| Weighted average |
> | FedTPG   | Text prompt generator  | No  | CLIP similarity  | Uniform averaging |
> | pFedPG   | Conditional prompt generator | No (client)    | CLIP similarity and conditions| Partial average |
> | FedAPT   | Adaptive prompt tuning | No | CLIP similarity  | Average  |
> | FedOTP   | Global and local prompts; Optimal transport | No (client)    | CLIP similarity  | Partial average |
>
> **Q2:** Experiments with more clients.
>
> **A2:** Thank you for your question. Our default setup indeed treats each domain as a single client (as in Tables 1 and 2). In the Appendix, we further split each domain’s data into five parts, thereby constructing 30 clients on DomainNet. The results, summarized in Table 6, show that our method still significantly outperforms the baselines under this setting. We will move these results to the main text in the revised version.
>
> **Q3:** The claim of reduced communication cost.
>
> **A3:** Thank you very much for your suggestion. We agree that our method is not the unique approach that can reduce the communication cost of CLIP-based federated learning, and we will clarify this point in the revision. Nevertheless, our communication advantage can be reflected by comparing the communication cost required to reach certain accuracy levels. In the table below, we report the number of communication rounds needed to achieve 20%, 40%, and 60% average accuracy on DomainNet. While the compared models have similar per-round communication costs, our method requires substantially fewer rounds in total, highlighting its communication efficiency.
>
> | acc. threshold | PromptFL  | FedCLIP | ADAPT |
> |-|:-:|:-:|:-:|
> | 20% | 24 | 35 |  9 |
> | 40% | 73 |   92 | 19 |
> | 60% | 121 | 147 | 57  |
>
> **Q4:** Comparison to other parameter-efficient techniques, e.g., LoRA.
>
> **A4:** Thanks for your feedback. Below we provide the comparison with LoRA. We emphasize that LoRA itself is merely a parameter-efficient fine-tuning technique and does not offer the convenient way that prompt learning provides to incorporate domain-aware mechanisms. Consequently, its predictive performance shows a significant gap compared to our method.
> |Method|clipart|infograph|painting|quickdraw|real|sketch|avg.|
> |-|:-:|:-:|:-:|:-:|:-:|:-:|:-:|
> |FLASC [1]|46.6|62.3|59.4 |32.7|80.6|67.4|58.2|
> |CLIP+LoRA (dual encoders)|73.8|49.2|69.7|33.5|81.5|59.4|61.2|
> |CLIP+LoRA (text side)|72.9|47.5|67.6|34.2|80.9|57.4|59.1|
> |Ours|78.5|64.7|72.5|44.5|85.8|73.2|69.8|
>
> [1] Kuo, Kevin, et al. Federated LoRA with Sparse Communication. CoRR 2024.
>
> **Q5:** Discussion of potential disadvantages.
>
> **A5:** Thank you for your advice. We believe that a major disadvantage and limitation of domain-specific federated learning is its reliance on sufficient data support, which may come either from large-scale pretraining data (e.g., CLIP’s training corpus) or from downstream fine-tuning datasets. When the data is insufficient or of low quality, the model struggles to overcome the convergence challenges introduced by domain gaps. For example, in our experiments, models without pretraining (such as FedAvg and FedProx) perform significantly worse than CLIP-based fine-tuning models, and in some cases, increasing the number of parameters even leads to a drop in accuracy. We will add this discussion in the revised version.
>
> **Q6:** Technical detail of optimizers.
>
> **A6:** Thanks for the question. We use SGD for ResNet and AdamW for ViT following standard training practices. Since ResNet was proposed earlier, most existing ResNet models were trained with SGD. After the introduction of ViT, the more advanced AdamW became the mainstream optimizer. We follow these practices to ensure the best performance of ViTs and fair comparisons with existing ResNets.
>
> **Q7:** Reporting standard deviation. **A7:** Thanks for the advice. We will present specific std numbers in the revised version.
>
> **Q8:** Formatting tables and figures. **A8:** Thank you for your recommendation. We will follow it to reorder the tables and figures.

---

### Review · Reviewer_Vcy8 · 2025-09-12

**Summary Of Contributions:**

The paper studies an important problem of prompt tuning and proposes a novel federated method. The method is sound and the paper is well-written. Experiments show the effectiveness of the proposed method.

**Audience:**

Yes

**Audience Explanation:**

This paper focuses on the prompt tuning, which is attractive to existing communities.

**Claims And Evidence:**

Yes

**Claims Explanation:**

The experiments on the baseline comparison, ablation study, and case studies show the effectiveness of the propsoed solutions.

**Requested Changes:**

1. It would be better to include more experiments regarding the parameter sensitity, which will provide more guidance for others about the parameter configuration.
2. It would also be better to provide more experiments in terms of efficiency, e.g., training time, test time, and the number of parameters.
3. It is suggested to discuss the future directions in the conclusion.

---

> ### Author Response · Authors · 2025-09-20
> **Author Response to Reviewer Vcy8**
>
> We appreciate your detailed feedback and constructive comments. Below is our detailed response to your concerns and questions.
>
> **Q1:** Parameter sensitivity.
>
> **A1:** Thank you for your advice. Below we present ablation results of our method with different training parameters. As shown, the accuracy remains within a reasonable fluctuation range. For example, even when the learning rate is increased by ten times, the average accuracy on DomainNet drops by only 2.4% compared to the default setting.
>
> | Learning Rate | Avg. Acc. (%) |
> |--------------|:---------------:|
> | 5e-5         | 66.9 |
> | 1e-4         | 67.2 |
> | **5e-4 (default)** | 69.8 |
> | 1e-3          | 69.1 |
> | 5e-3           | 67.4 |
>
> | Weight Decay | Avg. Acc. (%) |
> |-------------|:---------------:|
> | 0.001       | 69.0 |
> | 0.005        | 69.5 |
> | **0.01 (default)** | 69.8 |
> | 0.05         | 69.4 |
> | 0.1         | 68.8 |
>
> | Epochs | Avg. Acc. (%) |
> |-------|:---------------:|
> | 100    | 64.1 |
> | **200 (default)** | 69.8 |
> | 300    | 70.2 |
> | 400    | 70.4 |
> | 600    | 70.3 |
>
> **Q2:** Detailed efficiency comparison.
>
> **A2:** Thank you for your feedback. Below we provide a detailed comparison of training and inference throughput (images/second), and the number of learnable parameters for each client. The speed is tested on an A100 GPU. As shown the four listed methods share a similar level of training and inference speed, but our model has a clear advantage in performance.
>
> | Method      | Training throughput | Inference throughput | #Parameters (M) | Acc. (%) |
> |-------------|-|-|-|-|
> | FedAvg      | 175                | 676                          | 85.8         | 55.2|
> | PromptFL    | 81                 | 624                         | 0.08              | 63.2 |
> | FedCLIP     | 126                 | 572                         | 0.53             | 60.3 |
> | **Ours** | 80            | 578                          | 0.35              | 69.8 |
>
> **Q3:** Discuss the future directions in the conclusion.
>
> **A3:** Thanks. We will include related discussions in the revised version. Our main direction for future work is to scale up our algorithm for real-world applications, where clients may exhibit stronger domain gaps or even heterogeneous modalities. In such scenarios, our domain-aware prompt learning approach can provide effective insights for addressing these challenges.

---

### Review · Reviewer_X3s3 · 2025-09-15

**Summary Of Contributions:**

This paper proposes a way to finetune CLIP on domain-specific tasks via training a per-domain prompt in a federated way. When using non-i.i.d. data from different domains, the performance of traditional federated learning algorithms suffers, with even methods designed specifically for the heterogeneous setting showing much worse performance than Zero-shot CLIP (e.g. FedProx vs Zero-shot CLIP in Table 1). The paper answers this concern by developing ADAPT, a method that learns per-domain textual and visual prompts and where the text-vision components are coupled via AdaLN (which is just an affine transformation connection). Assuming that each client $i$ has data from domain $i$, ADAPT learns the prompts for domain $i$ using data from that client and shares it with the other clients for inference. Finally, the authors also introduce a momentum update to guard against performance drops from switching to using other clients' prompts after each update.

**Additional Comments:**

1. How would you modify your method to handle the case where the client-domain correspondence isn't perfect? e.g. let's say client 1 had data from domains 1,2 and client 2 had data from domains 2,3.
2. How about if each client had only one domain but it could be repeated? e.g. clients 2, 3 each have data from domain 2. Would you average the prompts? Would you use SGD with both?

**Audience:**

Yes

**Audience Explanation:**

I believe the paper is relevant to the literature on learning prompts for CLIP. I'm not really sure if it has wide applicability due to the limitations described above, but it could be useful down the line for the development of more sophisticated algorithms.

**Claims And Evidence:**

Yes

**Claims Explanation:**

1. The algorithm is quite simple, is shown to have good performance (Table 1) in the paper, and is explained well.
2. My main complaint is that the setting is quite narrow: we have $n$ clients with $n$ different domains neatly divided up, and I cannot really think of a realistic setting where this would be the case.
3. Data reconstruction is pretty difficult because there are so few parameters learned per client anyway.

**Requested Changes:**

Typos:
- "paining" in p.1

Claims:
- "However, fine-tuning the CLIP model may easily break the well-established alignment between vision and language, and CLIP will therefore lose the ability of open-vocabulary inference" can you provide a citation for this phenomenon?

Other:
- Will you share the code for this paper for reproducibility?

---

> ### Author Response · Authors · 2025-09-20
> **Author Response to Reviewer X3s3**
>
> We appreciate you thoughtful comments and constructive feedback. Below is the detailed response to your concerns and questions.
>
> **Q1:** Experimental setting is narrow. n clients with n different domains neatly divided up.
>
> **A1:** Thank you for your insightful comment. In fact, we considered two different settings in our experiments. One setting, used in the main experiments (Tables 1 and 2), treats each domain as a single client. The other setting further splits each domain into multiple clients, introducing both domain gap and label imbalance heterogeneity (Table 6 in the Appendix). In the revised version, we will reorganize the content and move the experiments with more clients into the main text. From the results, our method significantly outperforms the baselines under both settings, suggesting that our approach has strong generalization ability.
>
> **Q2:** Image reconstruction experiments.
>
> **A2:** Thank you for your feedback. We agree that the effectiveness of gradient-leakage-based image reconstruction is significantly affected by the number of trainable parameters, which implies higher security for parameter-efficient federated learning methods. However, this assumption has not been validated in the papers of the existing methods we compared against, whereas we are the first to provide such direct evidence. Our intention is not to claim this security property as a unique advantage of ADAPT, but rather to highlight, through direct comparison, the benefits of prompt-based federated learning methods over traditional approaches.
>
> **Q3:** Citation for breaking the alignment between vision language of CLIP.
>
> **A3:** Thank you for pointing out the missing citation. Here we load a CLIP image encoder from [1] which is fine-tuned on ImageNet and connect it with its original text encoder. Below we show its zero-shot classification results, where we can find it degrades severely compared to the original results. For some specific datasets like EuroSat, the accuracy is quite close to random guessing.
>
> | Dataset   | Food101 | EuroSAT | ImageNet | VOC2007 | SUN397 |
> |-----------|--------:|--------:|---------:|--------:|-------:|
> | CLIP ViT-B/16 (zero-shot) | 89.2 | 54.1 | 68.6 | 83.9 | 65.2 |
> | Finetuned CLIP ViT-B/16       | 23.9 | 12.3 | 32.2 | 29.4 | 20.8 |
>
>
> [1] Peng, et al. A Unified View of Masked Image Modeling
>
> **Q4:** Will you share the code for this paper for reproducibility?
>
> **A4:** Yes, our code, finetuned models, as well as the technical details for reproducibility will all be released after the paper is published.
>
> **Q5:** Handling complex client-domain correspondence.
>
> **A5:** Thank you for your insightful question. ADAPT is actually a highly flexible approach that can handle various federated learning scenarios. In the case where clients have overlapping domains, a simple modification is to allow each client to optimize all the prompts corresponding to the domains it possesses (whereas in our default setting, each client optimizes only one domain-specific prompt). Compared with our default non-overlapping setting, this scenario intuitively makes convergence easier since the data heterogeneity across clients is reduced. We will add discussions of these more complex scenarios in the revised version.
>
>
> **Q6:** Handling the scenario where multiple clients share a domain.
>
> **A6:** Thanks for your question. This setup is similar to the scenario mentioned in Q1. In this case, we aggregate the training parameters of clients belonging to the same domain. A detailed configuration is provided in Table 6 and the corresponding explanations in the Appendix (see Section Decentralization). Similar to our response to your previous question, ADAPT can be flexibly applied to different federated learning scenarios. The general principle is that each client only optimizes the prompt parameters corresponding to its visible domain, while parameters from the same domain are aggregated in a FedAvg-like manner.

---

### Decision · Action_Editor_7zJR · 2025-10-07

**Recommendation:** Accept with minor revision

**Additional Comments:**

There are substantial experiments in the rebuttal. The authors are encouraged to incorporate them into the revised paper.

**Audience:**

Yes

**Audience Explanation:**

Researchers working on domain adaptation and federated learning should be interested in this work.

**Claims And Evidence:**

Yes

**Claims Explanation:**

The experiments are supportive the effectiveness of the proposed prompt tuning, and the reviewers-authors discussion led to additional experiments about the baselines and parameter sensitivity.